# When Simple Problems Wear Complex Costumes: Improving Efficiency in LRM's Adaptive Reasoning

**Junnan Ren** [1]  **Yan Zhang** [1]  **Qian Chen** [2]  **Yunhang Shen** [3]  **Ke Li** [3]  **Shengchuan Zhang** [1]  **Liujuan Cao** [1]
**Rongrong Ji** [1 4]

## Abstract

Recent Large Reasoning Models (LRMs) have demonstrated powerful multi-step problem-solving capabilities but often suffer from inefficiency due to an "overthinking phenomenon", where they apply complex reasoning to simple tasks, resulting in unnecessary computational cost and latency. While adaptive reasoning models that can switch between generating explicit reasoning and producing direct answers offer a potential solution, their effectiveness is compromised by a critical flaw: they are often misled by superficial linguistic complexity, mistaking verbosely phrased simple problems for complex ones. To address this, we propose a two-stage training framework to create a more robust adaptive reasoner. The first stage uses supervised fine-tuning with augmented data—presenting simple problems in both concise and redundant forms—to teach the model to ignore superficial verbosity. Subsequently, a reinforcement learning phase utilizes Group Relative Policy Optimization (GRPO) with a custom reward function to refine the model's adaptive policy, ensuring it selects a reasoning mode based on true task complexity rather than surface-level cues. The resulting model reduces computational overhead without sacrificing accuracy and demonstrates improved robustness to misleading linguistic cues.

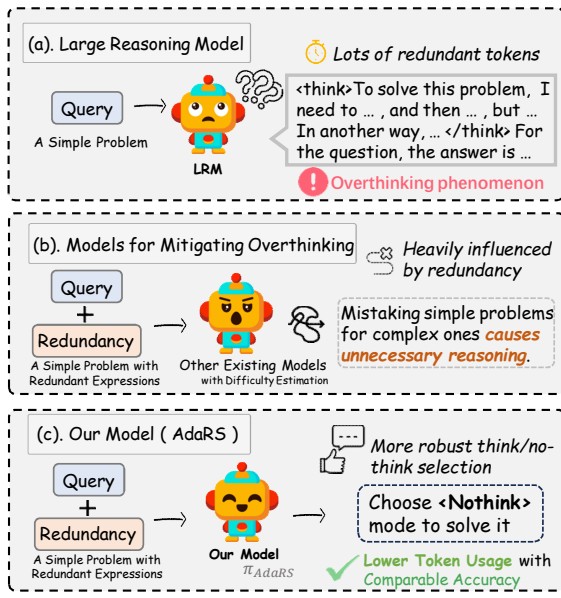

*Figure 1.* Conceptual illustration of the overthinking phenomenon and solutions. (a) A standard LRM produces verbose and unnecessary reasoning for simple problems. (b) Existing adaptive models may be misled by superficial linguistic complexity, mistakenly activating the Think mode. (c) Our model achieves more robust mode selection by distinguishing true problem complexity from redundancy.

## 1. Introduction

The recent emergence of Reasoning Large Language Models, including OpenAI o1 (OpenAI, 2024), DeepSeek-R1 (Guo et al., 2025), and others (Team et al., 2025), marks a significant leap forward. These models show a strong ability to solve complex tasks through multi-step reasoning. Powered by architectures and training approaches like reinforcement learning (Yu et al., 2022; Rafailov et al., 2023; Shao et al., 2024; Yu et al., 2025) and chain-of-thought techniques (Wei et al., 2022; Feng et al., 2023), these models address complex tasks by breaking them into intermediate logical steps, excelling in domains such as mathematics, computer programming, and science. However, the very mechanism underpinning their advanced performance–

---

[1]Key Laboratory of Multimedia Trusted Perception and Efficient Computing, Ministry of Education of China, Xiamen University, 361005, P.R. China. [2]School of Information Engineering, Xiamen Ocean Vocation College, Xiamen, 361102, China [3]Tencent Youtu Lab [4]Sino-Russian Research Center for Digital Economy. Correspondence to: Yan Zhang <bzhy986@xmu.edu.cn>.

namely, the generation of detailed, step-by-step reasoning chains–often gives rise to what is known as the 'overthinking phenomenon' (Chen et al., 2024b; Sui et al., 2025; Shen et al., 2025). This manifests as the production of redundant reasoning steps, especially when addressing problems of relatively low complexity which is illustrated in Figure 1.

To mitigate these efficiency challenges, researchers have explored various strategies, which predominantly focus on shortening the reasoning process itself (Zhou et al., 2024). These approaches, such as post-generation pruning or rewarding brevity, aim to reduce the token footprint of the reasoning chain (Xia et al., 2025; Yu et al., 2024; Cheng & Van Durme, 2024; Hao et al., 2024). However, a fundamental limitation of these approaches is that they still apply a reasoning process to every problem, failing to address instances where reasoning is entirely unnecessary.

More recently, a different paradigm has emerged that seeks to endow models with intrinsic adaptive reasoning capabilities (Liu et al., 2025; Lou et al., 2025). This approach equips models with two distinct modes: one for detailed step-by-step reasoning and another for direct answer generation, enabling dynamic switching to balance reasoning quality and computational cost (Jiang et al., 2025; Zhang et al., 2025a; Fang et al., 2025). This paradigm shows strong potential for improving reasoning efficiency, particularly by minimizing unnecessary computation on simpler tasks as illustrated in Figure 3. However, the efficacy of this paradigm hinges on the model's ability to reliably gauge a problem's intrinsic difficulty from its prompt, a mechanism whose robustness is underexplored. Our investigation uncovers a significant limitation in this area: models are often misled by superficial linguistic complexity, such as redundant phrasing, causing them to overestimate the difficulty of simple tasks and unnecessarily activate their effortful Think mode, which we illustrate in Figure 2. This observation motivates the development of adaptive reasoning models that focus on underlying problem complexity rather than superficial linguistic variations.

In this work, we introduce AdaReasoningSwitch (AdaRS), a model developed through a two-stage training framework. It is designed to be robust against superficial linguistic complexity, ensuring it activates the Think mode based on a problem's intrinsic difficulty rather than its surface phrasing, thereby improving reasoning efficiency.

The first stage uses supervised fine-tuning to establish two distinct modes, which we term the Think mode and the Nothink mode. The Think mode uses an analytical `<think>` format for complex tasks, while the Nothink mode employs a direct `<nothink>` format for simple ones. Crucially, to make the model robust to superficial linguistic complexity, we augment the training dataset with adversarial variants of simple problems—presenting each instance in both a concise form and in a version containing semantically neutral redundant expressions. This explicit pairing teaches the model that adding such irrelevant verbosity does not increase the underlying difficulty, and should therefore still be resolved via the efficient Nothink mode.

The second stage enhances this foundational skill through reinforcement learning. We employ Group Relative Policy Optimization (GRPO) (Shao et al., 2024) with a custom reward function to refine the model's decision-making policy. This function rewards the model for selecting the appropriate Think or Nothink mode based on true task complexity—which we define by a base model's ability to solve a problem without step-by-step reasoning—thereby training it to ignore superficial linguistic features.

We summarize our main contributions as follows: (1) To our knowledge, we are the first to systematically identify and demonstrate that current adaptive reasoning models are vulnerable to superficial linguistic complexity, often mistaking verbose phrasing for true problem difficulty. (2) We propose a two-stage training framework for AdaRS that enhances robustness to superficial linguistic cues, comprising supervised fine-tuning on data augmented with concise and redundant query variants, and reinforcement learning with a custom reward. This framework enables the model to distinguish true problem complexity from superficial linguistic redundancy. (3) We conduct comprehensive experiments showing that AdaRS achieves substantial reductions in computational cost without compromising accuracy. Moreover, evaluations on datasets with redundant phrasing show that AdaRS is more robust than existing methods, further supporting the effectiveness of our approach.

## 2. Related Work

LARGE REASONING MODELS.

Recent advancements have transformed large language models into powerful reasoning engines. This evolution was largely catalyzed by techniques like Chain-of-Thought (Wei et al., 2022) and its variants (Yao et al., 2023; Besta et al., 2024; Yang et al., 2024b), which guide models to break down complex problems by generating intermediate, step-by-step reasoning paths. Building on this, more sophisticated strategies have emerged to further enhance problem-solving accuracy, often by exploring multiple reasoning avenues (Zeng et al., 2024). Frontier systems, often referred to as Large Reasoning Models (LRMs) like OpenAI's o1, DeepSeek-AI's DeepSeek-R1, and Qwen's QwQ (OpenAI, 2024; Guo et al., 2025; Yang et al., 2025), exemplify this deep, human-like thinking (Wang et al., 2024). Their capabilities are typically cultivated through extensive reinforcement learning with outcome-based rewards or by fine-tuning on high-quality reasoning traces distilled from more capa-

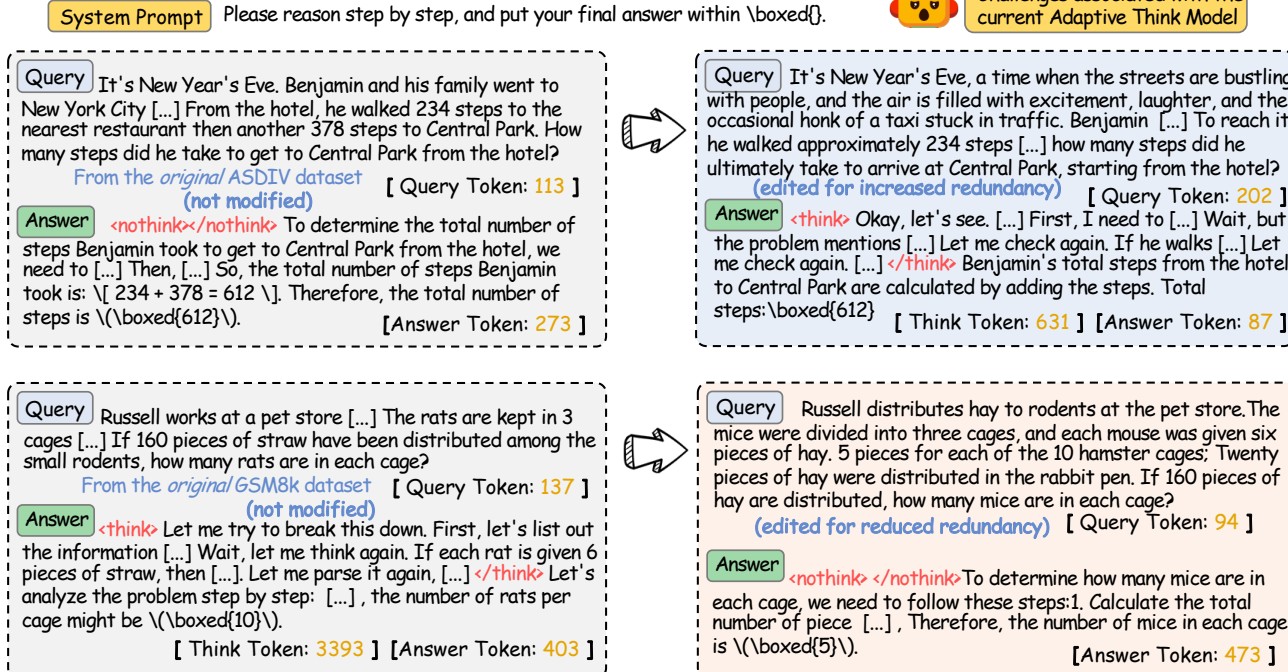

*Figure 2.* Demonstration of how superficial linguistic variations in queries affect the model's reasoning mode selection. For problems from two simple mathematical reasoning datasets, GSM8K and ASDIV, adding semantically neutral but redundant phrasing causes the model to incorrectly switch from the efficient 'nothink' mode to the resource-intensive 'think' mode, despite the underlying task simplicity remaining unchanged.

ble models (Tian et al., 2025). While these methods have pushed the state-of-the-art in performance on various reasoning benchmarks, their core reliance on generating long thought processes introduces a significant drawback: substantial inference latency and high computational costs. This inherent inefficiency has become a major bottleneck, motivating a dedicated field of research focused on improving the efficiency of reasoning in LRMs (Zhang et al., 2025b; Muennighoff et al., 2025; Chen et al., 2024a).

EFFICIENT REASONING.

The high computational cost of large reasoning models (LRMs) has motivated a range of approaches aimed at improving reasoning efficiency, which can be broadly grouped into two categories. The first and most common paradigm focuses on reducing the length and token footprint of the reasoning chain itself (Arora & Zanette, 2025; Yeo et al., 2025; Renze & Guven, 2024). This is achieved by techniques that adjust model behavior during or after generation, for example, using brevity-oriented rewards in reinforcement learning or fine-tuning on preference data favoring concise reasoning (Luo et al., 2025; Aggarwal & Welleck, 2025; Ma et al., 2025; Han et al., 2024). Alternatively, post-processing techniques can be used to condense verbose chains after they are generated. There also exist training-free methods that

use specialized prompting to encourage shorter responses without modifying the model's parameters (Lee et al., 2025; Xu et al., 2025). A limitation of all these methods, however, is that they still apply a reasoning process to every problem, failing to account for instances where it is entirely unnecessary.

A more recent paradigm, often termed Hybrid or Adaptive Reasoning, seeks to improve efficiency by deciding whether to engage in reasoning at all, dynamically switching between intensive and direct-answer modes (Jiang et al., 2025; Liu et al., 2025; Fang et al., 2025; Zhang et al., 2025a). This adaptability is realized either through multi-model systems, where a lightweight router directs queries, or within a single, unified model trained to support both behaviors. This method reduces computational cost by avoiding step-by-step reasoning on simple problems, without sacrificing performance on more complex ones. However, its reliance on surface cues remains a notable limitation, which we address by training the model to distinguish genuine complexity from superficial linguistic variation.

While these methods have laid the groundwork for adaptive reasoning, their effectiveness hinges on the critical and largely unexamined assumption that a model can accurately gauge a problem's intrinsic difficulty from its surface-level

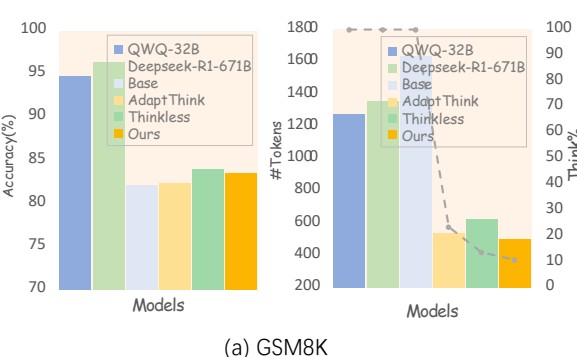
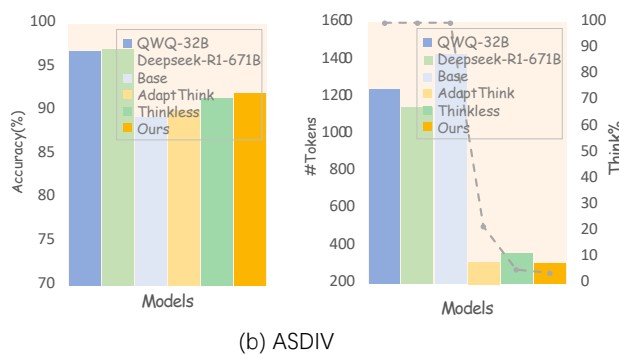

(a) GSM8K          (b) ASDIV

*Figure 3.* Performance comparison on the (a) GSM8K and (b) ASDIV datasets. The results demonstrate that, compared to strong reasoning models like QWQ-32B and Deepseek-R1-67B, adaptive reasoning models maintain comparable accuracy while substantially reducing token consumption by dynamically switching reasoning modes. The 'Base' model shown is DeepSeek-R1-Distill-Qwen-1.5B, which also serves as the foundation for the three adaptive models (Thinkless, AdaptThink, and our model, Ours). This efficiency gain is particularly pronounced on the simpler ASDIV dataset, where computational cost is significantly lowered without compromising performance.

presentation. Prior work has not sufficiently addressed the vulnerability of these systems to superficial linguistic variations, where verbose or elaborately phrased but simple problems can mislead a model into unnecessarily activating its costly reasoning mode. Our work directly confronts this challenge. We move beyond simply enabling a choice between modes and focus on making that choice robust, proposing a learning-based framework that trains the model to distinguish true problem complexity from superficial phrasal complexity.

## 3. Method

Our method focuses on optimizing a policy $\pi_\theta(o|q)$, parameterized by $\theta$, to generate an output sequence $o$ for a given query $q$. This policy implicitly selects between a Think or a Nothink reasoning mode by determining the initial tokens of the output sequence. The policy should maximize task accuracy while minimizing computational cost, particularly by avoiding the Think mode for problems that are simple in nature, regardless of their surface-level verbosity. To achieve this, we propose AdaReasoningSwitch, a model developed through the following two-stage training framework.

### 3.1. Cold Start with SFT

The initial stage of our framework employs supervised fine-tuning (SFT) to endow the base model, $\pi_{\text{init}}$, with the foundational capability for dual-mode reasoning. To achieve this, we construct a specialized SFT dataset by sourcing problems of varying intrinsic complexity. For problems requiring multi-step, complex reasoning, we create a Think subset, $\mathcal{D}_{\text{think}}$, where instances are formatted to guide the model to produce detailed reasoning steps encapsulated between `<think>` and `</think>` tokens, followed by the final answer. Conversely, for problems that are intrinsically simple, we create a Nothink subset, $\mathcal{D}_{\text{nothink}}$, where the model

is trained to generate the final answer directly following an empty `<nothink></nothink>` token pair, thereby bypassing the reasoning process. However, training solely on this initial dataset may cause the model to form spurious associations between verbose phrasing and task difficulty, leading it to mistakenly trigger the Think mode for simple problems.

To address this vulnerability, we introduce a targeted data augmentation strategy designed to make the model robust against superficial linguistic cues. We employ a powerful generator model, $\pi_{strong}$, to paraphrase the queries in our simple problem set, $\mathcal{D}_{\text{nothink}}$. For each query $q \in \mathcal{D}_{\text{nothink}}$, $\pi_{strong}$ generates two new variants: (1) a **redundant version** ($q_{\text{redundant}}$) that introduces superfluous phrasing without altering the problem's low intrinsic complexity, and (2) a **concise version** ($q_{\text{concise}}$) that strips the query to its minimal essential form. The concise variant is necessary because the original dataset may already contain a degree of incidental verbosity; by explicitly including a maximally concise formulation, we ensure the model observes the full spectrum—from minimal to highly verbose phrasing—all mapped to the same Nothink response, reinforcing that surface length is orthogonal to true difficulty. These augmented queries are paired with their original Nothink formatted answers, creating two additional data subsets: $\mathcal{D}_{\text{redundant}}$ and $\mathcal{D}_{\text{concise}}$. By explicitly training the model on these adversarial examples—where verbose queries still map to a Nothink response—we teach it to ignore verbosity. The final dataset for the supervised fine-tuning stage, $\mathcal{D}_{\text{SFT}}$, is therefore the union of the original and augmented data: $\mathcal{D}_{\text{SFT}} = \mathcal{D}_{\text{think}} \cup \mathcal{D}_{\text{nothink}} \cup \mathcal{D}_{\text{redundant}} \cup \mathcal{D}_{\text{concise}}$.

With the composite dataset fully assembled, the base model $\pi_{\text{init}}$ is fine-tuned on $\mathcal{D}_{\text{SFT}}$. The training objective is to minimize the standard language model cross-entropy loss, which aligns the model's policy with the expert demonstrations in

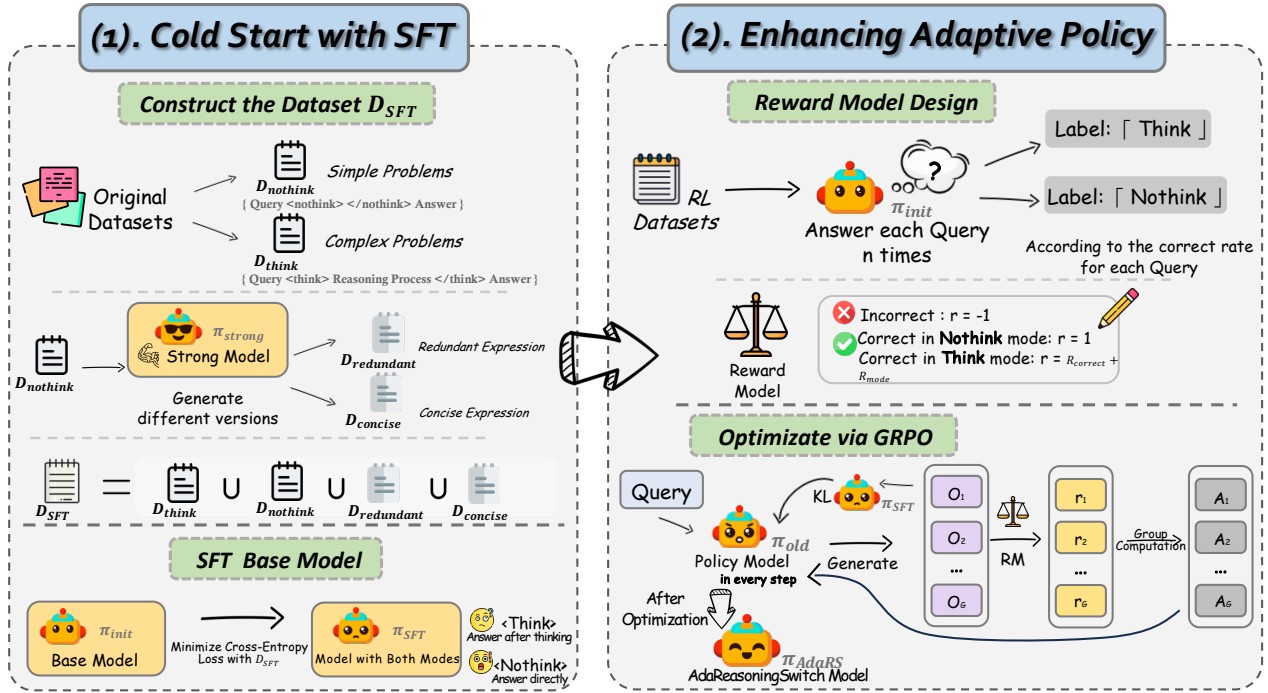

*Figure 4.* An overview of the two-stage training framework for AdaReasoningSwitch. (1) The "Cold Start with SFT" stage involves supervised fine-tuning on a dataset containing both concise and redundant problem variants, allowing the model to learn dual-mode (Think and Nothink) reasoning behavior. (2) The "Enhancing Adaptive Policy" stage employs Group Relative Policy Optimization (GRPO) with a custom reward function to refine the model's ability to select the appropriate reasoning mode based on true task complexity.

the dataset. The loss is defined as:

$$\mathcal{L}_{\text{SFT}}(\theta) = \mathbb{E}\left[-\sum_{t=1}^{T} \log \pi_\theta(o_t | q, o_{<t})\right], \qquad (1)$$

where $(q, o)$ is a training pair from $\mathcal{D}_{\text{SFT}}$, with $q$ as the input and $o = (o_1, \ldots, o_T)$ as the target sequence. The model's predicted probability $\pi_\theta(o_t | q, o_{<t})$ is defined over each token $o_t$ conditioned on the query and previous outputs. Training minimizes this loss to obtain $\pi_{\text{SFT}}$, which acquires dual-mode generation capability and serves as the starting point for reinforcement learning.

### 3.2. Enhancing Adaptive Policy

While the Cold Start stage enables $\pi_{SFT}$ to operate in both Think and Nothink modes, it does not optimize the decision policy between them. To instill this decision-making capability, we introduce a reinforcement learning phase using Group Relative Policy Optimization (GRPO). GRPO is well-suited for this task due to its effectiveness in complex reward landscapes, as our policy must learn to weigh computational cost against problem difficulty based on subtle linguistic cues.

The GRPO objective function is designed to improve the policy $\pi_\theta$ by maximizing a clipped surrogate objective, while

a KL-divergence penalty constrains the update to remain close to the reference policy $\pi_{\text{SFT}}$ for training stability. The objective is given by:

$$\mathcal{J}_{\text{GRPO}}(\theta) = \mathbb{E}\left[\frac{1}{G} \sum_{i=1}^{G} \left(L_i(\theta) - \beta \, \mathbb{D}_{\text{KL}}(\pi_\theta \| \pi_{\text{SFT}})\right)\right]. \quad (2)$$

Here, the expectation is over queries $q$ from our dataset and responses $o$ sampled from $\pi_{\text{old}}$, the policy from the previous training step. The term $L_i(\theta)$ is the clipped surrogate objective, defined as:

$$L_i(\theta) = \min\left(\rho_i(\theta) A_i, \, \text{clip}\left(\rho_i(\theta), \, 1 - \varepsilon, \, 1 + \varepsilon\right) A_i\right) \quad (3)$$

where $\rho_i(\theta) = \pi_\theta(o_i | q) / \pi_{\text{old}}(o_i | q)$ is the importance sampling ratio between the current and old policies, and $\varepsilon$ is a clipping hyperparameter. The advantage $A_i$, which estimates the relative value of a response, is calculated by standardizing the raw rewards within a batch of size $G$. This technique reduces variance and stabilizes the learning signal:

$$A_i = \frac{r_i - \text{mean}(\{r_1, r_2, \cdots, r_G\})}{\text{std}(\{r_1, r_2, \cdots, r_G\})}, \quad (4)$$

where $r_i$ is the scalar reward assigned to response $o_i$, calculated according to our custom reward function.

The effectiveness of the RL stage hinges on a reward function that aligns with our primary goal: maximizing accuracy while using the deliberative Think mode only when necessary. To achieve this, we designed a reward mechanism based on an empirical definition of problem complexity.

First, we establish a ground-truth label (`think` or `nothink`) for each question in our dataset. We define a question $\mathcal{Q}$ as "simple" if our base model $\pi_{\text{base}}$ can consistently answer it correctly without reasoning. Formally, this occurs if the model's accuracy in a forced `nothink` mode over $n$ trials exceeds a threshold $\tau$:

$$\frac{1}{n} \sum_{i=1}^{n} \mathbb{I}\big(\pi_{\text{base}}(\mathcal{Q}) = \text{GroundTruth}\big) \geq \tau, \qquad (5)$$

where $\mathbb{I}(\cdot)$ is the indicator function. Questions that do not meet this criterion are labeled `think`. We note that this definition is model-relative: a problem deemed "simple" here reflects the specific base model's capability rather than an absolute property of the problem itself. However, since our objective is to optimize this particular model's adaptive policy, this operational definition is well-aligned with our goal. With these labels, we define the reward $r$ to steer the policy's behavior:

**Incorrect Answer:** $r = -1$.

**Correct Answer with Nothink mode:** $r = +1$.

**Correct Answer with Think mode:** $r = R_{\text{correct}} + R_{\text{mode}}$.

where a correct answer produced in the Think mode receives a combined reward. Here, $R_{\text{correct}}$ denotes a fixed reward for producing the correct answer, while $R_{\text{mode}}$ is a positive bonus awarded only when the Think mode is appropriately used for a problem pre-labeled as `think`. By design, the maximum reward attainable via the Think mode ($R_{\text{correct}} + R_{\text{mode}}$) is set strictly lower than the Nothink reward ($+1$). This asymmetry creates a clear incentive hierarchy: the model is always better off choosing the efficient Nothink path whenever it can achieve correctness without reasoning. The Think mode becomes the rational choice only when direct answering would likely produce an incorrect result (reward $-1$), making the expected reward of engaging in deliberate reasoning higher despite its lower ceiling. In this way, the reward naturally steers the model to reserve costly reasoning exclusively for problems where it is the most reliable path to securing correctness.

## 4. Experiment

### 4.1. Experimental Setup

BASELINES.

To comprehensively evaluate the effectiveness of AdaRS, we compare it against three distinct categories of baselines:

(1) **General Performance Reference:** We utilize standard instruction-tuned (Qwen2.5-1.5B-Instruct) and math-tuned (Qwen2.5-Math-1.5B) models (Yang et al., 2024a) to establish a fundamental performance benchmark.

(2) **Backbone Variants and Manual Modes:** We assess our backbone model, DeepSeek-R1-Distill-Qwen-1.5B (Guo et al., 2025), across three specific configurations to isolate reasoning costs: a default Think mode; a Nothink mode where reasoning is programmatically bypassed by prepending `<think>\n\n</think>` control tokens; and a Random baseline that arbitrarily selects between these modes to benchmark non-adaptive performance. Additionally, we include the Qwen3 series (Yang et al., 2025), which integrates distinct modes via "Thinking Mode Fusion." However, unlike AdaRS, Qwen3 relies on manual user control for mode switching, serving as a reference for non-autonomous dual-mode architectures.

(3) **State-of-the-Art Efficient Reasoning:** We compare against advanced methods built on similar backbones, including chain-shortening approaches like OverThink (Liu et al., 2025), O1-Pruner (Luo et al., 2025), and CoT-Valve (Ma et al., 2025), as well as adaptive reasoning frameworks such as Thinkless (Fang et al., 2025) and AdaptThink (Zhang et al., 2025a).

EVALUATION METRICS.

We evaluated our model on a suite of mathematical reasoning benchmarks of increasing difficulty: ASDIV and GSM8K (elementary-level) (Miao et al., 2021), Math500 (high school competition) (Lightman et al., 2023), and AIME 2024 (advanced mathematics competition) (AI Mathematical Olympiad, 2024). To specifically test robustness against superficial linguistic complexity, we also created adversarial versions, GSM8K-redundancy and ASDIV-redundancy. Our primary evaluation metrics were Accuracy (pass@1) and Token Usage (computational efficiency). For models with adaptive capabilities, we additionally recorded Think%, indicating the activation rate of the Think mode, to analyze their adaptive reasoning behavior.

IMPLEMENTATION DETAILS.

Our implementation is based on the DeepSeek-R1-Distill-Qwen-1.5B model. All experiments ran on one node with eight A800 80G GPUs. (1) In the Cold Start stage, we built a composite dataset to train the model for dual-mode rea-

*Table 1.* Main performance comparison on three benchmarks. We report both Accuracy (%) ↑ and the average number of tokens consumed per instance (#Tokens) ↓. Among the efficient variants, the best result is bolded and the second-best is underlined.

| Models | GSM8K | | Math500 | | AIME 2024 | | Average | |
|---|---|---|---|---|---|---|---|---|
| | Accuracy | #Tokens | Accuracy | #Tokens | Accuracy | #Tokens | Accuracy | #Tokens |
| **Large Language Models:** | | | | | | | | |
| Qwen2.5-1.5B-Instruct | 72.4 | 475 | 58.4 | 747 | 9.3 | 1267 | 46.7 | 830 |
| Qwen2.5-Math-1.5B | 78.6 | 453 | 79.6 | 684 | 12.1 | 1391 | 56.8 | 842.7 |
| **Large Reasoning Models:** | | | | | | | | |
| DeepSeek-R1-Distill-Qwen-1.5B$_{Think}$ | 82.1 | 1634 | 80.1 | 4929 | 28.1 | 15675 | 62.4 | 7413 |
| DeepSeek-R1-Distill-Qwen-1.5B$_{Nothink}$ | 71.2 | 304 | 69.4 | 802 | 16.1 | 2488 | 52.2 | 1198 |
| **Efficient Reasoning Models:** | | | | | | | | |
| OverThink | 77.2 | 709 | 81.2 | 4131 | 28.3 | 11269 | 62.3 | 5369 |
| O1-Pruner | 74.8 | **458** | 82.2 | 3212 | **28.9** | 10361 | 61.9 | 4677 |
| CoT-Valve | 79.7 | 1009 | 80.2 | 5167 | 19.3 | 17245 | 59.7 | 7807 |
| Thinkless | **84.0** | 621 | 81.2 | 2488 | 23.0 | 7305 | 62.7 | 3471 |
| AdaptThink | 82.3 | 540 | 83.9 | 2481 | 27.0 | 8778 | 64.4 | 3933 |
| AdaRS (ours) | 83.5 | 503 | **84.4** | **2293** | **28.9** | **6954** | **65.6** | **3250** |

*Table 2.* Performance and adaptive behavior on a specialized MMLU benchmark.

| Models | Accuracy | #Tokens | Think% |
|---|---|---|---|
| Qwen3-0.6B$_{Think}$ | 59.6 | 3385 | 100% |
| Qwen3-0.6B$_{Nothink}$ | 39.5 | 582 | 0% |
| Qwen3-1.7B$_{Think}$ | 81.2 | 3976 | 100% |
| Qwen3-1.7B$_{Nothink}$ | 57.1 | 872 | 0% |
| Qwen3-4B$_{Think}$ | 86.9 | 3272 | 100% |
| Qwen3-4B$_{Nothink}$ | 73.5 | 814 | 0% |
| DeepSeek-R1-Distill-Qwen-1.5B$_{Think}$ | 59.0 | 1901 | 100% |
| DeepSeek-R1-Distill-Qwen-1.5B$_{Nothink}$ | 48.1 | 691 | 0% |
| Thinkless | 65.1 | 1310 | 56.4% |
| AdaptThink | 46.9 | 998 | 77.9% |
| AdaRS(ours) | 66.0 | 1172 | 45.3% |

soning. It combines OpenR1-Math-220k for Think mode and a subset of Hybrid-OpenThoughts2-1M for Nothink mode. To improve robustness against superficial linguistic complexity, this dataset was augmented with adversarial query variants (both redundant and concise) generated by a powerful generator model $\pi_{strong}$. We fine-tuned the model for 2 epochs using a learning rate of 1e-5 and cosine decay. (2) We optimized the model's adaptive policy using the GRPO algorithm on the DeepScaleR-Preview-Dataset. The reward function assigned $R_{correct} = 0.2$ and a bonus $R_{mode} = 0.3$ for correctly selecting Think mode on complex problems—defined as those where the base model failed to exceed 80% accuracy in 10 Nothink trials($\tau = 0.8, n = 10$). Key GRPO hyperparameters included batch size $G = 8$, KL coefficient $\beta$ of 0.001 for training stability, and learning rate of 1e-6. We capped the generation length at 16K and selected the final checkpoint after 350 steps based on validation performance. Further details on data processing and experimental setup are provided in the Appendix.

## 4.2. Main Results and Analysis

As demonstrated in Table 1, AdaRS achieves a favorable balance between accuracy and computational cost across benchmarks of varying difficulty. When compared to its base model, DeepSeek-R1-Distill-Qwen-1.5B (Think mode), AdaRS improves average accuracy from 62.4% to 65.6% while substantially reducing average token consumption by 56.1% (from 7413 to 3250 tokens). Notably, this efficiency is achieved while maintaining competitive performance on the most challenging AIME 2024 benchmark (28.9% accuracy). Furthermore, the marked improvement in accuracy over the forced Nothink base mode (52.2%) highlights the necessity of on-demand reasoning, allowing the model to address complex problems without compromising efficiency on simpler tasks. Compared to other adaptive reasoning models such as Thinkless and AdaptThink, AdaRS achieves the highest average accuracy and the lowest token usage, suggesting that its reinforcement learning phase promotes a more nuanced understanding of when reasoning is needed. In contrast to methods like O1-Pruner that truncate the reasoning process and often degrade performance, AdaRS maintains high accuracy. This demonstrates the effectiveness of our adaptive approach in preserving core reasoning abilities while improving efficiency.

To evaluate the generalization capabilities of our model on broader reasoning tasks, we benchmarked its performance on four relevant subjects from the MMLU dataset, as shown in Table 2. The results for the non-adaptive Qwen3 series highlight the trade-off between accuracy and computational efficiency. For instance, the Qwen3-4B model achieves high accuracy in Think mode, while consuming 3272 tokens. Conversely, switching to Nothink mode substantially reduces token usage to 814 tokens but results in a significant drop in accuracy. Due to the absence of an

*Table 3.* Analysis of adaptive reasoning policies on simple datasets and their versions with added redundancy. The table reports average token consumption (#Tokens) ↓ and the activation rate of the Think mode (Think%) ↓ . Across all adaptive models, the best result in each column is bolded and the second-best is underlined.

| Models | GSM8K | | ASDIV | | GSM8K$_{redundancy}$ | | ASDIV$_{redundancy}$ | |
|---|---|---|---|---|---|---|---|---|
| | #Tokens | Think% | #Tokens | Think% | #Tokens | Think% | #Tokens | Think% |
| Qwen3-0.6B$_{Think}$ | 2225 | 100% | 1234 | 100% | 2198 | 100% | 1916 | 100% |
| Qwen3-0.6B$_{Nothink}$ | 379 | 0% | 293 | 0% | 696 | 0% | 662 | 0% |
| Qwen3-1.7B$_{Think}$ | 2205 | 100% | 1395 | 100% | 2591 | 100% | 2130 | 100% |
| Qwen3-1.7B$_{Nothink}$ | 409 | 0% | 289 | 0% | 774 | 0% | 727 | 0% |
| Qwen3-4B$_{Think}$ | 2215 | 100% | 1533 | 100% | 2586 | 100% | 2201 | 100% |
| Qwen3-4B$_{Nothink}$ | 405 | 0% | 295 | 0% | 780 | 0% | 735 | 0% |
| DeepSeek-R1-Distill-Qwen-1.5B$_{Think}$ | 1634 | 100% | 1430 | 100% | 2154 | 100% | 2319 | 100% |
| DeepSeek-R1-Distill-Qwen-1.5B$_{Nothink}$ | 304 | 0% | 237 | 0% | 732 | 0% | 751 | 0% |
| DeepSeek-R1-Distill-Qwen-1.5B$_{Random}$ | 972 | 48.9% | 839 | 50.1% | 1375 | 53.5% | 1608 | 49.8% |
| Thinkless | 621 | 13.7% | 362 | 5.5% | 1050 | 42.3% | 1356 | 40.7% |
| AdaptThink | 540 | 23.6% | 318 | 21.9% | 1512 | 65.4% | **1081** | 58.5% |
| AdaRS(ours) | **503** | **10.9%** | **308** | **4.3%** | **874** | **35.9%** | 1123 | **33.1%** |

*Table 4.* Ablation study on the components of our framework. The table shows the incremental gains from the Cold Start stage and highlights the advantage of our Enhancing Adaptive Policy over GRPO with a naive reward.

| Models | GSM8K | | ASDIV | |
|---|---|---|---|---|
| | Accuracy | #Token | Accuracy | #Token |
| DeepSeek-R1-Distill-Qwen-1.5B | 82.1 | 1634 | 89.3 | 1430 |
| + Cold Start | 82.7 | 1047 | 91.8 | 869 |
| + GRPO with Naive Reward | 82.9 | 1809 | 91.4 | 1766 |
| + Enhancing Adaptive Policy | **83.5** | **503** | **92.1** | **308** |

autonomous switching mechanism, these models necessitate manual decisions that frequently involve suboptimal trade-offs between accuracy and efficiency, highlighting the importance of adaptive system design. Adaptive models based on the DeepSeek-R1-Distill-Qwen-1.5B backbone highlight the value of dynamic mode selection. AdaRS achieves the highest accuracy with the lowest think-mode activation, demonstrating strong reasoning selectivity. In contrast, AdaptThink activates think mode most frequently but yields the lowest accuracy , indicating inefficient reasoning use. These results validate the effectiveness of AdaRS and the robustness of our two-stage training framework.

To directly test the core assumption of this work, namely the model's robustness to superficial linguistic complexity, we conducted a comparative experiment on both the standard and adversarially-generated redundant versions of the GSM8K and ASDIV datasets. As shown in Table 3, the results reveal a critical vulnerability in existing adaptive models. The Random variant serves as a baseline that selects Think or Nothink mode with equal probability per query. When presented with redundant phrasing, Thinkless and AdaptThink exhibit a sharp increase in Think mode activation; for instance, AdaptThink's activation on ASDIV-redundancy rises to 58.5% from 21.9% on the original dataset. This confirms that their policies often mistake

linguistic verbosity for true problem complexity. In contrast, while AdaRS is not entirely immune to these linguistic cues, it demonstrates considerably greater robustness. Although its Think activation rate also increases, its final activation rate on the redundant datasets remains lower than its counterparts. This indicates that our training framework successfully mitigates, though does not completely eliminate, the model's sensitivity to superficial phrasing. This residual sensitivity suggests an avenue for future work in more advanced adversarial data augmentation.

As shown in Table 4, our ablation study validates the effectiveness of our two-stage framework. The initial Cold Start stage, by itself, provides a substantial efficiency gain, reducing token consumption, while also slightly improving accuracy. This establishes the value of dual-mode training even before policy optimization. This highlights the importance of our reward design: using GRPO with a naive reward that ignores reasoning cost causes the model to overuse the Think mode, significantly increasing token consumption. In contrast, our final Enhancing Adaptive Policy, which incorporates a reward for appropriate mode selection, achieves the highest accuracy while drastically reducing token costs on both datasets. This confirms that both the dual-mode SFT and the carefully designed RL policy are essential for achieving optimal performance and efficiency.

## 5. Discussion

The widespread "overthinking" problem in Large Reasoning Models (LRMs) limits both their efficiency and broader adoption. While previous work has mainly focused on post-training optimization such as reinforcement learning or instruction fine-tuning, it often overlooks a core issue: how does a model decide when to engage in reasoning and when to respond directly? Evidence shows that models tend to rely on superficial heuristic cues, including input length,

technical terms, or prompt phrasing like "please reason step by step." These cues can unnecessarily trigger complex reasoning, even for tasks that require only basic logic or a single-step calculation. This behavior reveals a key weakness in current models—they are proficient in pattern recognition but lack the ability to assess true task complexity. Addressing this issue requires more than shortening reasoning chains or toggling between modes. Instead, it calls for training models to distinguish between superficial linguistic complexity and genuine logical difficulty.

## 6. Conclusion

We introduced a novel training framework to mitigate adaptive models' sensitivity to superficial linguistic cues. It first applies contrastive fine-tuning on syntactically varied yet semantically equivalent queries, guiding the model to attend to underlying complexity over surface phrasing. This capability is further refined through GRPO with a custom reward, aligning the model's decisions with actual task complexity. The resulting model shows greater robustness and strikes a better balance between accuracy and efficiency, significantly reducing token usage by avoiding needless reasoning on simple tasks.

## Acknowledgements

This work was supported by the National Key Research and Development Program of China (No.2025YFE0113500), the National Science Fund for Distinguished Young Scholars (No.62525605), and the National Natural Science Foundation of China (No. U25B2066, No.62506313, No.U22B2051).

## Impact Statement

This paper presents work whose goal is to advance the field of Machine Learning. There are many potential societal consequences of our work, none which we feel must be specifically highlighted here.

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

# Appendix

# A. Details of Experiments

## Baselines

The baseline models fall into three main categories: standard instruction-tuned models, large reasoning models, and advanced efficient reasoning methods. Below, we detail the key models used in our comparisons.

The DeepSeek-R1-Distill-Qwen-1.5B model serves as the foundational backbone for our experiments and is a critical baseline for comparison. To thoroughly evaluate the spectrum of reasoning efficiency, we assess this model in several configurations. Its standard Think mode represents the default behavior, where the model generates a comprehensive, step-by-step reasoning chain before arriving at an answer. To create a direct-answering counterpart, the Nothink mode is simulated by prepending the model's input with `<think>\n\n</think>` control tokens, forcing it to bypass the reasoning process entirely. In addition to these fixed modes, we also established a Random version as a non-adaptive baseline. For each query, this version randomly selects whether to use the Think or Nothink mode. This configuration provides a crucial probabilistic benchmark, allowing us to measure the performance of our intelligent policy against a simple, non-adaptive chance of engaging in costly reasoning.

The Qwen3 series is included as a key baseline because it is one of the first models to feature inherently distinct Think and Nothink modes, which are integrated through a post-training phase called "Thinking Mode Fusion". This provides a valuable benchmark for the potential of a dual-mode architecture. However, a crucial limitation of the Qwen3 series is that the switching between these two modes is not adaptive; it must be manually controlled by the user for each query. This lack of an autonomous decision-making mechanism means that while the model possesses the capability for both complex reasoning and efficient direct answers, it cannot intelligently select the appropriate mode on its own, highlighting the need for an adaptive system like AdaRS.

Details on other baselines such as standard instruction-tuned models (Qwen2.5-1.5B-Instruct), math-tuned models (Qwen2.5-Math-1.5B), and other efficient or adaptive reasoning models (OverThink, O1-Pruner, CoT-Valve, Thinkless, and AdaptThink) are covered in the main experimental setup (Section 4.1).

## Datasets

TRAINING DATASETS.

Our two-stage training process uses the following datasets. OpenR1-Math-220k provides 220k math problems for Think mode SFT with step-by-step reasoning examples. A subset of Hybrid-OpenThoughts2-1M was used for Nothink mode SFT, training direct-answer generation. DeepScaleR-Preview-Dataset contains 40k math problems used during the RL phase to refine the model's adaptive policy.

EVALUATION DATASETS.

We evaluated our model across benchmarks of increasing difficulty. GSM8K (8,500 problems) and ASDIV (2,305 problems) are elementary-level math word problem benchmarks. Math500 is a curated set of 500 high-school competition problems spanning seven subjects from the MATH dataset. AIME 2024 contains 15 advanced competition-level problems from the American Invitational Mathematics Exam. GSM8K-redundancy and ASDIV-redundancy are adversarial versions of the elementary benchmarks, created by paraphrasing original problems with semantically neutral but superfluous language using $\pi_{strong}$, to test robustness against superficial linguistic complexity. For MMLU, we selected four subjects—Abstract Algebra, College Computer Science, Formal Logic, and High School Mathematics—to evaluate generalization beyond mathematical reasoning (detailed per-subject results in Table 5).

| Models | Abstract Algebra | | College Computer Science | | Formal Logic | | High School Mathematics | |
|---|---|---|---|---|---|---|---|---|
| | Accuracy | #Tokens | Accuracy | #Tokens | Accuracy | #Tokens | Accuracy | #Tokens |
| Qwen3-0.6B$_{Think}$ | 53.1 | 3108 | 55.3 | 3276 | 51.6 | 3199 | 78.5 | 3883 |
| Qwen3-0.6B$_{Nothink}$ | 37.3 | 578 | 40.0 | 539 | 45.2 | 583 | 35.6 | 629 |
| Qwen3-1.7B$_{Think}$ | 84.1 | 3736 | 72.0 | 5047 | 81.8 | 3659 | 87.0 | 3463 |
| Qwen3-1.7B$_{Nothink}$ | 52.3 | 730 | 59.0 | 1038 | 59.5 | 750 | 57.8 | 970 |
| Qwen3-4B$_{Think}$ | 92.9 | 2868 | 80.2 | 3728 | 86.0 | 2851 | 88.5 | 3639 |
| Qwen3-4B$_{Nothink}$ | 79.4 | 823 | 76.0 | 814 | 76.2 | 772 | 62.2 | 845 |
| DeepSeek-R1-Distill-Qwen-1.5B$_{Think}$ | 50.0 | 1344 | 61.1 | 2234 | 46.8 | 1704 | 78.1 | 2320 |
| DeepSeek-R1-Distill-Qwen-1.5B$_{Nothink}$ | 36.1 | 649 | 54.5 | 734 | 40.9 | 689 | 60.6 | 693 |
| Thinkless | **60.4** | 1881 | 62.0 | 1435 | 58.7 | 1749 | 79.6 | 1774 |
| AdaptThink | 43.7 | **771** | 46.9 | 1939 | 41.3 | **743** | 55.7 | **1233** |
| AdaRS(ours) | 58.9 | 1011 | **64.1** | **1134** | **61.1** | 1230 | **79.8** | 1312 |

*Table 5.* Detailed test results of four subjects in MMLU.

```
# Role
You are a text transformation tool. Your only function is to rewrite a given word problem into a more verbose and narratively complex version.
# Task
I will provide a simple English word problem. You will rewrite it by adding irrelevant context, distracting details, and a narrative background. The core mathematical problem must remain unchanged. You must adhere to all rules, especially the output format rule.
# Rules
1. **Preserve Core Logic:** The essential numbers and the mathematical operation required to solve the problem (e.g., division, multiplication) must be the same as in the original problem.
2. **Expand Existing Context:** Take the scenario from the original problem and build upon it. Add details about the characters, the setting, the reasons behind the actions, and other sensory information.
3. **Inject Irrelevant "Noise":** Integrate new numbers, secondary characters, or minor events that have no bearing on the final answer. This is the primary method for adding redundancy.
4. **Single Paragraph Output:** The final output must be a single, coherent paragraph that flows naturally as one long question.
5. **Strict Output Format:** **CRITICAL RULE.** Your response must contain *only* the text of the new, verbose question. Do not output *any* other text. Do not write "Here is the generated question:", "Output:", or any other form of introduction or explanation. Your entire response must be the generated problem itself.
# Example
*   **Input Problem:**
    `Peter has 54 tickets and 4 pencils. If he shares the tickets among 9 friends, how many tickets does each friend get?`
*   **Your Required Output (Exact Format):**
    `Peter had a great birthday party last Saturday at the Galaxy Arcade, filled with noisy pinball machines and flashy VR games. He hit the jackpot on one game and won 54 prize tickets, plus 4 colorful pencils he decided to ignore. Back at school on Monday, he wants to fairly share the tickets with his 9 best friends who came to the party. If he divides them evenly, how many tickets does each friend get?`
# Your Task
Rewrite the following problem according to all the rules above. Remember the strict output format.
```

*Figure 5.* An example of the prompt given to our generator model, $\pi_{strong}$, to increase the superficial linguistic complexity of a problem. The goal was to generate a verbose query without altering the underlying mathematical task, thereby creating adversarial examples to test robustness.

### A.1. Implementation Details

Our implementation is based on the DeepSeek-R1-Distill-Qwen-1.5B model, with all experiments conducted on a single node equipped with eight A800 80G GPUs. The training process was divided into two main stages. The first stage, Supervised Fine-Tuning (SFT), was implemented using the LLaMA-Factory framework for its efficiency in large-scale model training. The second stage, Reinforcement Learning (RL), was built on the Verl framework, a library specialized for RL with large language models.

The initial SFT stage focused on endowing the model with dual-mode (Think and Nothink) reasoning capabilities. To achieve this, we constructed a composite dataset. The Think portion consisted of problems from OpenR1-Math-220k. For the Nothink portion, we selected a subset of simple questions from Hybrid-OpenThoughts2-1M. To enhance robustness against superficial linguistic complexity, we then used the generator model $\pi_{strong}$ to generate both redundant and concise versions for each of these simple questions. This augmented set, combined with the Think data, formed the final SFT dataset. We fine-tuned the model on this composite dataset for 2 epochs using a learning rate of 1e-5 with a cosine decay schedule for stable convergence.

In the second stage, we used the Group Relative Policy Optimization (GRPO) algorithm to refine the model's adaptive policy. This phase utilized the DeepScaleR-Preview-Dataset, of which approximately 15,000 examples were identified as simple problems. For the remaining more difficult problems, we established a ground-truth complexity label: a problem was labeled as requiring the *Think* mode if the base model failed to exceed 80% accuracy over 10 forced *Nothink* trials ($\tau = 0.8$, $n = 10$). The reward function assigned $R_{correct} = 0.2$ for a correct answer and a bonus of $R_{mode} = 0.3$ for correctly selecting the *Think* mode for these complex problems. Key GRPO hyperparameters included a batch size $G = 8$, a KL coefficient $\beta$ of $0.001$ for training stability, and a learning rate of $1e-6$. The maximum generation length was capped at 16,000 tokens, and the final model checkpoint was selected after 350 training steps based on validation performance.

```
# Task
I will provide an English word problem. First, you must evaluate if it is already concise. If it is, return the original problem without changes. If the problem is verbose, you will rewrite
it to be as brief as possible while retaining the core mathematical problem. You must adhere to all rules, especially the output format rule.
# Rules
1. **Assess for Conciseness:** If the problem already contains only the essential information needed for mathematical calculation, it should be considered concise, and you must
output the original text.
2. **Preserve Core Logic:** The essential numbers and the mathematical operation required to solve the problem (e.g., division, multiplication) must be the same as in the original
problem.
3. **Remove Redundancy:** Delete all irrelevant background, distracting details, narrative descriptions, and sensory information. Focus only on the data needed for the solution.
4. **Filter "Noise":** Eliminate any secondary numbers, characters, or minor events that have no bearing on the final answer.
5. **Single Paragraph Output:** The final output must be a single, coherent paragraph.
6. **Strict Output Format:** **CRITICAL RULE.** Your response must contain *only* the text of the new, concise question. Do not output *any* other text. Do not write "Here is
the generated question:", "Output:", or any other form of introduction or explanation. Your entire response must be the generated problem itself.
# Example 1: Simplification Needed
*   **Input Problem:**
    `Peter had a great birthday party last Saturday at the Galaxy Arcade, filled with noisy pinball machines and flashy VR games. He hit the jackpot on one game and won 54 prize
tickets, plus 4 colorful pencils he decided to ignore. Back at school on Monday, he wants to fairly share the tickets with his 9 best friends who came to the party. If he divides them
evenly, how many tickets does each friend get?`
*   **Your Required Output (Exact Format):**
    `Peter has 54 tickets. If he shares the tickets among 9 friends, how many tickets does each friend get?`
# Example 2: Already Concise, No Change Needed
*   **Input Problem:**
    `A farmer has 15 apples and sells 7 of them. How many apples does the farmer have left?`
*   **Your Required Output (Exact Format):**
    `A farmer has 15 apples and sells 7 of them. How many apples does the farmer have left?`
# Your Task
Rewrite the following problem according to all the rules above. Remember the strict output format.
```

*Figure 6.* An example prompt used with our generator model, $\pi_{strong}$, to produce a concise version of a problem. Unlike the redundancy prompt, it includes logic to leave already simple problems unchanged. The goal is to generate concise counterparts to redundant queries, creating data pairs that help the model learn that core difficulty is independent of surface phrasing.

## B. Prompt Design

To construct the augmented dataset for training, we carefully designed prompts to guide the generator model ($\pi_{strong}$) in creating both redundant and concise versions of simple problems. These prompts were aimed at generating samples for adversarial training, thereby teaching our model to distinguish between superficial linguistic complexity and true problem difficulty. An example of the prompt used to generate verbose and complex problems is shown in Figure 5, while the prompt for generating concise versions is shown in Figure 6.

## C. Case Study

To specifically demonstrate the effectiveness of our AdaRS model, we provide a case study that directly compares the responses of our model with the baseline model on a simple problem. This case highlights the "overthinking" phenomenon present in the baseline model, which engages in unnecessarily verbose reasoning for a simple task. In contrast, our AdaRS model accurately assesses the problem's intrinsic simplicity, thereby selecting the more efficient Nothink mode. A detailed comparison for this case is presented in Figure 7.

## D. Further Discussion and Limitations

While our work demonstrates the effectiveness of adaptive reasoning mode selection, several limitations warrant discussion.

**Binary mode selection.** Our current framework treats reasoning as a binary decision: the model either engages in full deliberative reasoning (Think) or bypasses it entirely (Nothink). In practice, problem complexity exists on a spectrum—some problems may benefit from brief, shallow reasoning (e.g., a few verification steps) rather than either exhaustive chain-of-thought or no reasoning at all. Exploring a finer-grained, multi-level reasoning paradigm that allows the model to calibrate the depth of its reasoning to match varying degrees of problem complexity is a promising direction for future work.

**Simplified redundancy modeling.** The redundant query variants used in our training and evaluation are generated by a language model, which may not fully capture the diversity of real-world linguistic redundancy. In practice, users introduce verbosity through a wide range of natural behaviors—habitual filler phrases, colloquial digressions, repetitive clarifications, or domain-specific jargon—each of which may affect a model's mode selection in different ways. Moreover, since both the augmented data and the reward signal are derived from a narrow set of redundancy patterns, there is a risk that the model overfits to these specific patterns rather than learning a truly general robustness, potentially leading to reward hacking. A systematic study of how different types and sources of real-world redundancy influence adaptive reasoning behavior, along with more diverse augmentation strategies, remains an important direction for future investigation.

**Unified model burden and architectural alternatives.** Our framework places the entire burden of both difficulty assessment and response generation on a single model. In contrast, modern agentic systems and multi-model pipelines

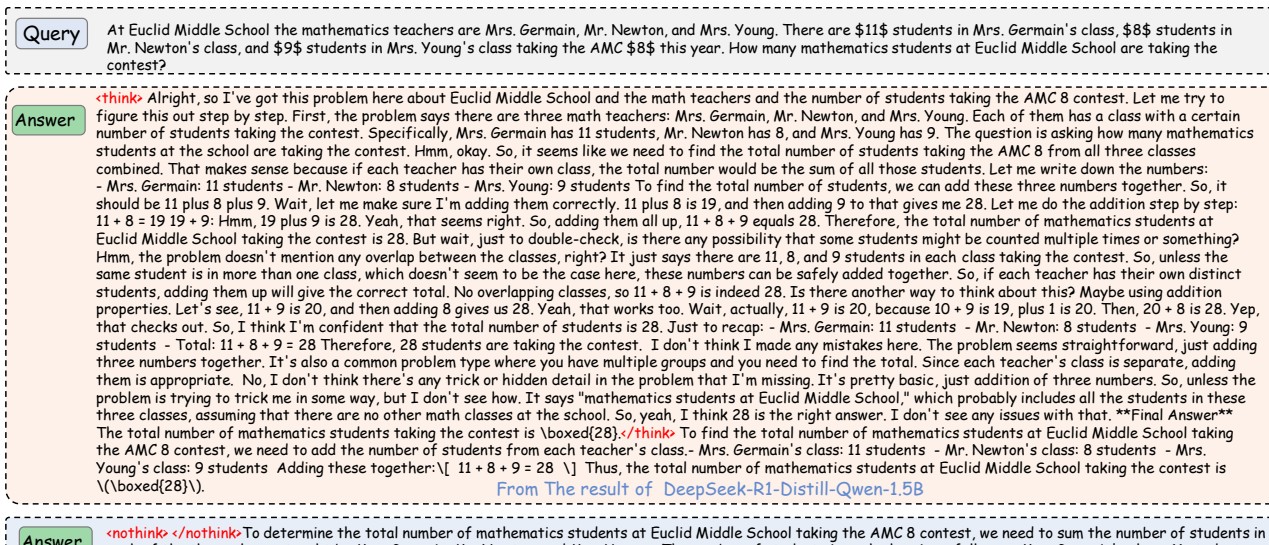

*Figure 7.* A case study comparing the responses of the base model (DeepSeek-R1-Distill-Qwen-1.5B) and our AdaRS model to a simple mathematical query. The base model unnecessarily activates its Think mode, leading to verbose reasoning, whereas AdaRS correctly identifies the problem's simplicity and uses the efficient Nothink mode to provide a direct answer.

often employ a lightweight external router to triage queries before dispatching them to specialized models. Such architectures can decouple the complexity judgment from the reasoning process, potentially offering more flexible and scalable adaptive strategies. Investigating how our adaptive training framework interacts with or could benefit from these multi-model routing paradigms is a valuable avenue for future work.

**Model-centric difficulty assessment.** Our approach relies entirely on the model's own capability to assess problem complexity. However, in many real-world deployment scenarios—particularly on-device or user-facing applications—the appropriate level of reasoning effort may depend not only on intrinsic problem difficulty but also on user intent, task context, or domain-specific requirements. For instance, a user may prefer a quick approximate answer over a precise but slow one, or a specific application may require categorizing queries into fine-grained instruction types with corresponding response strategies. Incorporating human preferences or external signals into the mode selection policy remains an open challenge.

**Scope of evaluation.** Our experiments are conducted exclusively on a 1.5B-parameter model and focus primarily on mathematical reasoning tasks. While these settings allow controlled comparisons, the generalizability of our findings to larger model scales (e.g., 7B or above) and to other reasoning domains such as code generation, commonsense reasoning, or multi-hop question answering has not been verified.

**Residual sensitivity to linguistic cues.** As shown in our robustness experiments (Table 3), AdaRS significantly reduces but does not fully eliminate sensitivity to superficial linguistic complexity. The Think mode activation rate still increases when faced with redundantly phrased problems, suggesting that more advanced adversarial training strategies or architectural modifications may be needed to achieve full robustness.

