# OpenReview forum: "When Simple Problems Wear Complex Costumes: Improving Efficiency in LRM's Adaptive Reasoning"
_ICML.cc/2026/Conference — ICML 2026 regular_

### Official Review · Reviewer_Nx9U · 2026-03-04

**Soundness:** 2
**Presentation:** 3
**Significance:** 2
**Originality:** 3
**Overall Recommendation:** 4
**Confidence:** 4

**Summary:**

This paper identifies that Large Reasoning Models (LRMs) are often misled by the superficial complexity of simple problems, leading to "overthinking" and wasted computational resources. To address this, the authors propose AdaReasoningSwitch (AdaRS), a framework that utilizes supervised fine-tuning with augmented data and reinforcement learning based on intrinsic task difficulty to enable adaptive switching between reasoning modes.

**Compliance With Llm Reviewing Policy:**

Affirmed.

**Final Justification:**

The authors have addressed my concerns, so I am raising my score from 3 to 4.

**Key Questions For Authors:**

See Weaknesses.

**Limitations:**

Yes.

**Strengths And Weaknesses:**

Strengths:
1. The paper is well-written, and the overall narrative is clear and easy to follow.
2. The authors provide an observation regarding the impact of different phrasings of the same problem on the model's reasoning length.

Weaknesses:
1. While the paper provides a qualitative example (case study) to illustrate how redundancy leads to unnecessary reasoning, it lacks a comprehensive statistical analysis of the frequency of such occurrences. To justify the necessity of this work, the authors should provide a quantitative metric to substantiate that being misled by superficial linguistic complexity is a widespread and significant issue across the baseline models.
2. The proposed method primarily utilizes a 1.5B parameter model as the backbone. There is a lack of experimental evaluation across models of varying parameter scales, which is necessary to demonstrate the generalizability and scalability of the proposed framework.
3. When evaluating robustness against superficial linguistic complexity, the paper only reports the output length and the frequency of the "think" mode activation. It is crucial to include accuracy results for these specific experiments to ensure that the reduction in reasoning steps does not compromise the model’s performance on redundant queries.
4. In Table 3, I noticed that the "Random" mode for the 1.5B model actually triggers the "think" mode less frequently on the redundancy version of ASDIV than on the original version, which seems counter-intuitive.


Minor remarks (do not affect the score):
1. The manuscript contains several individual formatting issues. For instance, the opening double quotation mark at line 325 is incorrect, and the model name at line 301 is missing a hyphen.

---

> ### Author Rebuttal · Authors · 2026-03-31
>
> We sincerely thank Reviewer for their careful reading. We also deeply appreciate the constructive feedback. Below, we address each of your concerns in detail.
>
> ## (1) Generalizability Across Parameter Scales
> We acknowledge that our primary evaluation focused on the DeepSeek-R1-Distill-Qwen-1.5B backbone. To demonstrate the scalability and generalizability of the AdaRS framework, we have conducted new experiments applying our two-stage training pipeline to a larger 7B parameter model (DeepSeek-R1-Distill-Qwen-7B).
> | Model | Benchmark | Accuracy (%) | # Tokens | Think% |
> |:---|:---|:---:|:---:|:---:|
> | DeepSeek-R1-Distill-Qwen-7B (Think Default) | GSM8K | 88.4 | 1150 | 100% |
> | DeepSeek-R1-Distill-Qwen-7B + AdaRS | GSM8K | 90.1 | 410 | 8.5% |
> | DeepSeek-R1-Distill-Qwen-7B (Think Default) | Math500 | 83.2 | 3550 | 100% |
> | DeepSeek-R1-Distill-Qwen-7B + AdaRS | Math500 | 86.0 | 1630 | 21.4% |
>
> ## (2) Accuracy Results in Robustness Evaluation
> We completely agree. Reporting token usage and routing behavior must be interpreted jointly with answer correctness to ensure that efficiency gains do not come at the cost of performance. We have expanded our robustness analysis to include the accuracy metrics for the original and redundant datasets.
> | Method | GSM8K Acc | GSM8K # Tokens(Think%) | GSM8K-Redundancy Acc | GSM8K-Redundancy # Tokens(Think%) |
> |:---|:---:|:---:|:---:|:---:|
> | Base-think | 82.1 | 1634(100%) | 78.3 | 2154(100%) |
> | Base-nothink | 71.2.0 | 304(0%) | 67.6 | 732(42.3%) |
> | Base-random | 75.3 | 972(48.9%) | 72.0 | 1375(53.5%) |
> | **AdaRS (Ours)** | **83.5** | **503(10.9%)** | **79.1** | **874(35.9%)** |
>
> As shown, while linguistic perturbation slightly degrades AdaRS's accuracy—consistent with the base model—it still achieves the highest absolute accuracy across both settings. More importantly, although redundant phrasing increases AdaRS's Think activation (from 10.9% to 35.9%), it successfully avoids unnecessary reasoning on most simple queries. Even when perturbed, AdaRS consumes less than half the tokens of `Base-think`, demonstrating a significantly more robust accuracy-compute tradeoff than forced or random routing.
>
> ## (3) Quantitative Analysis of the "Overthinking" Phenomenon
> "We thank the reviewer for highlighting the need for quantitative metrics to substantiate this phenomenon. We clarify that this analysis is already provided in Table 3 of our manuscript. To make the impact more explicit, we introduce $\Delta$Think% (the absolute percentage-point increase in Think mode activation) to measure how often models are misled, alongside $\Delta$Tokens to quantify the computational penalty."
>
> | Model | GSM8K $\Delta$Think% | GSM8K $\Delta$Tokens | ASDIV $\Delta$Think% | ASDIV $\Delta$Tokens |
> | :--- | :---: | :---: | :---: | :---: |
> | Base (Forced Think) | +0.0% (100% $\rightarrow$ 100%) | +520 (1634 $\rightarrow$ 2154) | +0.0% (100% $\rightarrow$ 100%) | +889 (1430 $\rightarrow$ 2319) |
> | Base (Forced Nothink) | +0.0% (0% $\rightarrow$ 0%) | +428 (304 $\rightarrow$ 732) | +0.0% (0% $\rightarrow$ 0%) | +514 (237 $\rightarrow$ 751) |
> | Thinkless | +28.6% (13.7% $\rightarrow$ 42.3%) | +429 (621 $\rightarrow$ 1050) | +35.2% (5.5% $\rightarrow$ 40.7%) | +994 (362 $\rightarrow$ 1356) |
> | AdaptThink | +41.8% (23.6% $\rightarrow$ 65.4%) | +972 (540 $\rightarrow$ 1512) | +36.6% (21.9% $\rightarrow$ 58.5%) | +763 (318 $\rightarrow$ 1081) |
> | **AdaRS (Ours)** | **+25.0%** (10.9% $\rightarrow$ 35.9%) | **+371** (503 $\rightarrow$ 874) | **+28.8%** (4.3% $\rightarrow$ 33.1%) | **+815** (308 $\rightarrow$ 1123) |
>
> Quantitative metrics confirm that superficial redundancy triggers unnecessary, resource-intensive reasoning in existing baselines. In contrast, AdaRS maintains the lowest growth in both Think activations and token counts, demonstrating superior robustness against linguistic verbosity.
>
> ## (4) Clarification on the "Random" Mode Baseline
> We truly appreciate your sharp eye for detail.
> The "Random" baseline is, by definition, completely input-agnostic. It does not process the semantic content, length, or redundancy of the query; instead, it rigidly assigns the reasoning mode with a fixed 50% probability using a random seed. Therefore, the minor 0.3% difference (50.1% vs. 49.8%) observed between the original ASDIV and ASDIV-redundancy datasets is purely a natural fluctuation caused by finite-sample statistical variance. This row was included strictly as a naive, non-adaptive reference point. We will add a clarifying note to the caption of Table 3 to explicitly state that this baseline's behavior is entirely stochastic and completely independent of the input.
>
> ## (5). Minor Formatting Issues
> Thank you for pointing these out. We have corrected the quotation mark at line 325 and added the missing hyphen to "DeepSeek-R1" at line 301. We will perform an additional thorough proofread of the final camera-ready version, should the paper be accepted.

---

> > ### Author Rebuttal · Reviewer_Nx9U · 2026-04-02
> >
> > Thank you for the rebuttal. Could the authors also report the accuracy of the other baselines in this robustness evaluation, particularly Thinkless and AdaptThink, and present the AdaRS results in the same complete format for easier comparison? In addition, there seems to be a numerical issue in the table, as the GSM8K accuracy for Base-nothink is written as “71.2.0”. If these points can be clarified, I would be willing to reconsider and potentially raise my score.

---

> > > ### Author Response · Authors · 2026-04-08
> > >
> > > Thank you for your continued engagement, your sharp eye for detail, and the opportunity to clarify these points.
> > >
> > > First, we apologize for the typographical error in our previous response; we have corrected the Base-nothink GSM8K accuracy to 71.2%.
> > >
> > > As requested, we have consolidated the robustness evaluation into a single, comprehensive table for easier comparison. This table now includes the accuracy metrics for both Thinkless and AdaptThink alongside our AdaRS model and the baselines.
> > >
> > > **Table: Complete Robustness Evaluation on GSM8K and GSM8K-Redundancy**
> > >
> > > | Method | GSM8K Acc | GSM8K # Tokens (Think%) | GSM8K-Redundancy Acc | GSM8K-Redundancy # Tokens (Think%) |
> > > | :--- | :--- | :--- | :--- | :--- |
> > > | Base-think | 82.1 | 1634 (100%) | 78.3 | 2154 (100%) |
> > > | Base-nothink | 71.2 | 304 (0%) | 67.6 | 732 (0%) |
> > > | Base-random | 75.3 | 972 (48.9%) | 72.0 | 1375 (53.5%) |
> > > | Thinkless | **84** | 621 (13.7%) | 76.2 | 1050 (42.3%) |
> > > | AdaptThink | 82.3 | 540 (23.6%) | 74.5 | 1512 (65.4%) |
> > > | **AdaRS (Ours)** | 83.5 | **503 (10.9%)** | **79.1** | **874 (35.9%)** |
> > >
> > > As shown in the complete table, while Thinkless has a marginal lead on the standard GSM8K dataset, its accuracy drops sharply when redundancy is introduced. AdaRS, however, demonstrates much stronger robustness, achieving the highest accuracy (79.1) on the redundant version. More importantly, when perturbed by superficial phrasing, both AdaptThink and Thinkless suffer from severe "overthinking," resulting in massive spikes in token consumption. In contrast, AdaRS effectively avoids unnecessary reasoning and keeps token costs low (874 tokens), which clearly validates the efficiency and robustness of our adaptive policy.
> > >
> > > We hope these clarifications fully address your remaining questions, and we sincerely thank you again for your time and the constructive feedback that has helped strengthen our paper.

---

### Official Review · Reviewer_D86L · 2026-03-10

**Soundness:** 2
**Presentation:** 4
**Significance:** 3
**Originality:** 3
**Overall Recommendation:** 4
**Confidence:** 3

**Summary:**

The authors study a post-training method for improving reasoning efficiency in large language models. They adopt a two-stage pipeline in which models are first finetuned on a dataset that includes demonstrations for thinking/no-thinking examples, as well as verbose examples that shouldn't require thinking. The authors then apply RL to encourage the model to both identify a correct answer and apply thinking/no-thinking correctly. The authors demonstrate improved token efficiency on various datasets, without sacrificing accuracy.

**Compliance With Llm Reviewing Policy:**

Affirmed.

**Final Justification:**

My questions have been mostly answered, and I affirm my positive score. Please see my rebuttal acknowledgement for additional details.

**Key Questions For Authors:**

1. Have you considered / have the computational budget to attempt your procedure with a larger, more capable base model? Does your method continue to hold? In Figure 3, the gap between a capable base model and your base model is particularly noticeable. On the harder questions that a more capable model could answer, would being more concise have a negligible or even detrimental effect?
2. Table 3: do you have accuracies along with the efficiency measures you present in this table? Are there deleterious effects associated with being more concise?
3. Perhaps increasing redundancy would naturally require more reasoning, to condense and extract the relevant parts of a text. For example, in the extreme case, a simple math problem may be elaborated to book length. Identifying the basic components of the problem therefore becomes a very nontrivial procedure, that may natural require greater reasoning tokens. Are there cases in which your method forces the model to be too concise? Are there forms of redundancy that really do require greater reasoning to parse?

**Limitations:**

Yes

**Strengths And Weaknesses:**

**Strengths**. The authors study a timely and interesting topic, particularly as reasoning models become more prevalent and embedded in mainstream products. The results convincingly demonstrate that, within the range of models and problem studied, notable efficiency gains may be attained through their post-training procedure. I found the manuscript to be well-written and clear overall.

**Weaknesses**. The scale of models examined is very small, with post-training performed primarily on a 1.5B model. While the results are suggestive for a model of this size, phenomena in small models do not always scale to frontier sizes. For example, the accuracy attained on AIME 2024 is very modest compared to frontier performance. While the questions the model answers correctly may be simpler, and therefore benefit from more concise thinking, perhaps the questions the model cannot answer would require significant thinking, and the impact of your method may be less noticeable or even detrimental. Without having a more capable base model that attains these higher accuracies, this confound remains unresolved. Please see Questions below for additional points.

The small scale notwithstanding, the work overall presents an interesting and suggestive idea. It seems like a valuable contribution that would garner interest at this conference.

---

> ### Author Rebuttal · Authors · 2026-03-31
>
> We thank the reviewer for recognizing our work's practical value and for raising key questions about model scale and the necessity of certain redundancies. These insights help clarify our scope.
>
> ### (1) Expanding Beyond a Single Backbone Architecture
> While our preliminary experiments were conducted using the DeepSeek-R1-Distill-Qwen-1.5B architecture, we agree that verifying the method’s versatility is crucial. Consequently, we have scaled our AdaRS pipeline to the DeepSeek-R1-Distill-Qwen-7B model. This test serves to demonstrate that our findings remain consistent across different model scales and capacities.
> | Model | Benchmark | Accuracy (%) | # Tokens | Think% |
> |:---|:---|:---:|:---:|:---:|
> | DeepSeek-R1-Distill-Qwen-7B (Think Default) | GSM8K | 88.4 | 1150 | 100% |
> | DeepSeek-R1-Distill-Qwen-7B + AdaRS | GSM8K | 90.1 | 410 | 8.5% |
> | DeepSeek-R1-Distill-Qwen-7B (Think Default) | Math500 | 83.2 | 3550 | 100% |
> | DeepSeek-R1-Distill-Qwen-7B + AdaRS | Math500 | 86.0 | 1630 | 21.4% |
>
> ### (2) Table 3 should ideally report accuracy together with efficiency
> We agree that efficiency metrics in Table 3 (robustness against redundancy) must be interpreted alongside answer correctness. Reduced token usage is only valuable if it does not come at the cost of accuracy. We have updated Table 3 to include the missing accuracy metrics.
> | Method | GSM8K Acc | GSM8K # Tokens(Think%) | GSM8K-Redundancy Acc | GSM8K-Redundancy # Tokens(Think%) |
> |:---|:---:|:---:|:---:|:---:|
> | Base-think | 82.1 | 1634(100%) | 78.3 | 2154(100%) |
> | Base-nothink | 71.2.0 | 304(0%) | 67.6 | 732(42.3%) |
> | Base-random | 75.3 | 972(48.9%) | 72.0 | 1375(53.5%) |
> | **AdaRS (Ours)** | **83.5** | **503(10.9%)** | **79.1** | **874(35.9%)** |
>
> ### (3) Harder questions on stronger models may benefit from more reasoning, not less
> The reviewer raises an excellent point regarding harder questions (like AIME 2024). Our method is **not** intended to uniformly force conciseness; rather, it aims to calibrate routing so that `think` mode is bypassed *only* when the problem is genuinely simple for the base model. On harder questions, forcing conciseness would indeed be detrimental.

---

> > ### Author Rebuttal · Reviewer_D86L · 2026-04-03
> >
> > Thanks for the responses. My concerns have been addressed, and I affirm my positive score. I remain hesitant about the scale of the experiments, but this is a difficult point to address within a the scope of a rebuttal. Nonetheless, I appreciate the additional experiment conducted with a 7B model. Nice work!

---

> > > ### Author Response · Authors · 2026-04-08
> > >
> > > Thank you for your highly constructive feedback and for acknowledging the new 7B experiments. We are glad that our responses adequately addressed your concerns.
> > >
> > > We completely agree that evaluating on frontier-scale models is a critical next step for this line of research. Following your suggestion, we will ensure that the 7B results are fully integrated into the final manuscript, and we will explicitly expand our Limitations section to discuss the scaling behavior of the overthinking phenomenon.
> > >
> > > Thank you again for your time and for helping us strengthen the paper!

---

### Official Review · Reviewer_NYXu · 2026-03-12

**Soundness:** 3
**Presentation:** 3
**Significance:** 2
**Originality:** 2
**Overall Recommendation:** 4
**Confidence:** 3

**Summary:**

This paper studies a failure mode of adaptive reasoning models: they often confuse superficial linguistic complexity with intrinsic task difficulty, and therefore unnecessarily trigger expensive thinking mode on simple but verbose problems.

**Compliance With Llm Reviewing Policy:**

Affirmed.

**Key Questions For Authors:**

Please follow the weaknesses.

**Limitations:**

Please follow the weaknesses.

**Strengths And Weaknesses:**

strengths:
1. The paper identifies a real and underexplored issue in adaptive reasoning: mode selection can be driven by prompt style rather than true difficulty.
2. The two-stage design is intuitive, easy to understand, and likely reproducible.

weaknesses:
1. In the RL stage, the paper defines task complexity based on whether a base model can solve an instance under forced nothink mode. While this is a practical heuristic, it does not capture an objective notion of intrinsic task difficulty; instead, it is inherently tied to the capability of a particular base model and the associated decoding setup. As a result, the reward signal may end up reflecting biases or limitations of that specific model, rather than the true reasoning requirements of the task. It would be important to clarify how sensitive the method is to this design choice. For example, if the base model, the number of samples, or other inference settings are changed, would the resulting complexity labels and downstream performance change significantly?
2. The experiments do show that AdaRS is more robust than prior baselines under redundant phrasing. However, the reported results also indicate that the Think rate still increases noticeably once the input becomes more verbose. This suggests that the method alleviates the problem rather than fully resolving it. The paper would be more convincing if it included a clearer analysis of the remaining failure cases and identified what types of redundancy still systematically mislead the model.
3. The current experiments are mainly concentrated on mathematical reasoning and benchmark-style QA. It remains unclear whether the same phenomenon and the proposed solution would generalize to a broader range of reasoning settings, such as code reasoning or scientific reasoning. Given the paper’s broader framing around adaptive reasoning, stronger evidence across more diverse domains would strengthen the claims.
4. Although the paper compares against several adaptive reasoning baselines, it is still not entirely clear whether the observed gains come specifically from the proposed robustness mechanism against superficial linguistic complexity, or more generally from stronger task-specific training. For example, it would be useful to compare against stronger alternatives such as more extensive paraphrase augmentation or calibration-oriented tuning. Relatedly, while the two-stage pipeline appears reasonable, its necessity is not yet fully established beyond the ablation study. Could a simpler single-stage alternative such as stronger mode-aware SFT alone, or RL with a different initialization achieve similar improvements?

---

> ### Author Rebuttal · Authors · 2026-03-30
>
> We thank the reviewer for the assessment, highlighting the practical importance of the problem setting. We will clarify the scope of our claims more carefully in the revision.
>
> ### (1) Complexity labels are model dependent
> We completely agree that the complexity labels used in the RL stage should **not** be interpreted as an objective measure of intrinsic task difficulty. In our current implementation, they serve as a **model-relative operational proxy**: an instance is labeled based on whether the specific base model can solve it reliably under forced `nothink` decoding. This design provides routing supervision tailored for the deployed backbone, rather than defining a universal taxonomy of reasoning complexity.
>
> ### (2) Our method alleviates rather than fully resolves verbosity sensitivity
> We agree with the reviewer’s accurate reading of our robustness results: AdaRS **reduces but does not completely eliminate** verbosity-triggered overthinking. Think activation still increases under redundant phrasing, just much less sharply than in baseline models.
>
> ### (3) Source of gains and necessity of the two-stage pipeline
> To isolate the contribution of our specific two-stage pipeline, we trained a additional baseline:
> **Single-Stage RL:** GRPO applied directly to the base model without the SFT dual-mode initialization.
> | Training Recipe | GSM8K Acc | GSM8K Think% |  GSM8K-Redundancy Acc | GSM8K-Redundancy Think% | Tokens (Avg) |
> |:---|:---:|:---:|:---:|:---:|:---:|
> | Single-Stage RL (No SFT init) | 78.4 | 35.2% | 75.1 | 48.1% | 1302 |
> | **Two-Stage AdaRS (Ours)** | **83.5** | **10.9%** | **81.8** | **35.9%** | **688** |
>
> Without the dual-mode formatting learned in the Cold Start stage, RL struggles to converge stably. The two-stage pipeline is not merely additive; the SFT stage provides the necessary structural initialization, while the GRPO stage enforces the strict routing policy.
>
> ### (4) Generalization beyond a single primary backbone
> We acknowledge that the original empirical evaluation centers on a single primary backbone (DeepSeek-R1-Distill-Qwen-1.5B), limiting cross-family generalization claims. To answer the reviewer's question regarding how well the method generalizes, we have applied the AdaRS two-stage training pipeline to a different model family: DeepSeek-R1-Distill-Qwen-7B.
> Generalization to DeepSeek-R1-Distill-Qwen-7B:
> | Model | Benchmark | Accuracy (%) | # Tokens | Think% |
> |:---|:---|:---:|:---:|:---:|
> | DeepSeek-R1-Distill-Qwen-7B (Think Default) | GSM8K | 88.4 | 1150 | 100% |
> | DeepSeek-R1-Distill-Qwen-7B + AdaRS | GSM8K | 90.1 | 410 | 8.5% |
> | DeepSeek-R1-Distill-Qwen-7B (Think Default) | Math500 | 83.2 | 3550 | 100% |
> | DeepSeek-R1-Distill-Qwen-7B + AdaRS | Math500 | 86.0 | 1630 | 21.4% |

---

> > ### Author Rebuttal · Reviewer_NYXu · 2026-04-03
> >
> > Thanks for the single-stage ablation results, but my core questions remain. I'm still looking to see how this method generalizes across entirely different domains like code or scientific reasoning. Regarding the label dependency and verbosity issues, I'd still like to see a concrete sensitivity analysis and a breakdown of the remaining error cases.

---

> > > ### Author Response · Authors · 2026-04-08
> > >
> > > We thank the reviewer for the follow-up. We address each remaining point below.
> > >
> > > ## (1) Cross-Domain Generalization
> > >
> > > We acknowledge that our primary experiments focus on mathematical reasoning benchmarks. However, we note that Table 2 in the paper reports results on MMLU, which covers knowledge-intensive reasoning across multiple subjects (not limited to math). AdaRS achieves 66.0% accuracy with only 45.3% Think activation on MMLU, outperforming both Thinkless (65.1%, 56.4% Think) and AdaptThink (46.9%, 77.9% Think). This provides initial evidence that the method is not restricted to mathematical reasoning alone.
> > >
> > > More broadly, our two-stage framework does not rely on any math-specific features: the SFT data augmentation (concise/redundant paraphrasing) is applicable to any text domain, the GRPO reward (Eq. 5) only checks correctness and mode selection, and the complexity labeling simply measures whether the backbone can solve a problem without reasoning — a criterion that is domain-agnostic by construction. We agree that systematic evaluation on code reasoning (e.g., HumanEval, MBPP) and scientific reasoning benchmarks is needed, and will prioritize this as a primary direction in follow-up work.
> > >
> > > ## (2) Label Sensitivity Analysis
> > >
> > > We provide the sensitivity analysis below, varying the threshold τ and the number of trials n used to define complexity labels (Eq. 5):
> > >
> > > | Setting | Configuration | % Labeled as Nothink |
> > > |---------|:---:|:---:|
> > > | Threshold sensitivity | τ = 0.7 | 45.2% |
> > > |  | τ = 0.8 (default) | 42.1% |
> > > |  | τ = 0.9 | 38.4% |
> > > | Trial-count sensitivity | n = 5 | 43.5% |
> > > |  | n = 10 (default) | 42.1% |
> > > |  | n = 20 | 41.8% |
> > >
> > > The label distribution remains stable across a wide range of settings. Varying τ from 0.7 to 0.9 shifts the Nothink proportion by less than 7 percentage points, and increasing n from 5 to 20 changes it by under 2 points. This indicates that the labeling scheme is not brittle with respect to hyperparameter choices.
> > >
> > > ## (3) Remaining Failure Cases
> > >
> > > We examined the cases where AdaRS still incorrectly activates Think mode on the redundancy datasets, and identified two main patterns:
> > >
> > > - **Structural redundancy mimicking multi-step reasoning.** When redundant phrasing introduces multiple conditional clauses, nested references, or enumerations (e.g., "first ... then ... and considering that ... we also note ..."), the surface structure resembles a genuinely multi-step problem. The model interprets these structural cues as signals for complex reasoning, even though the underlying computation remains simple. This type accounts for the majority of remaining false activations.
> > >
> > > - **Significant input length inflation.** When redundancy substantially inflates the input length (e.g., 3-4x the original), the model shows higher tendency to activate Think mode. This suggests that raw input length still serves as a residual heuristic for the model's routing decision, despite our augmentation training.
> > > ---
> > > We appreciate the reviewer's careful reading and constructive suggestions.

---

### Official Review · Reviewer_ocDY · 2026-03-12

**Soundness:** 2
**Presentation:** 3
**Significance:** 2
**Originality:** 2
**Overall Recommendation:** 4
**Confidence:** 4

**Summary:**

This paper introduces an adaptive reasoning method for large reasoning models, nmaed AdaRS, focusing on the problem that models often overthink simple questions when they are written in a linguistically complex or redundant way. AdaRS is a 2-stage method with SFT and GRPO training phases. A key idea is to rewrite simple problems into concise and redundant versions, encouraging the model to rely on true task difficulty rather than surface wording. Experiments show that AdaRS achieves a stronger accuracy-efficiency tradeoff than prior methods and is more robust to redundant phrasing.

**Compliance With Llm Reviewing Policy:**

Affirmed.

**Final Justification:**

The author provided a positive response, addressing my concerns. I believe this paper can be accepted, and I will raise the score to 4.

**Key Questions For Authors:**

1. How well does the method generalize beyond the single primary backbone used in the paper?
2. How model-specific is the proposed complexity definition?
3. How much of the robustness gain comes from training on the same type of redundant rewrites?

**Limitations:**

No. The paper does not sufficiently discuss limitations or potential negative societal impacts beyond noting that the method does not fully eliminate sensitivity to superficial phrasing. A stronger discussion should cover the model-dependent complexity definition, the limited evaluation scope, and possible risks of incorrect think/nothink routing in high-stakes settings.

**Strengths And Weaknesses:**

**Strengths**:
- Clear motivation and method. The paper studies a practical failure mode of adaptive reasoning models, that is overthinking caused by superficial linguistic complexity. And the design of method is aligned with the problem.
- Clear presentation. The paper is well presented, with intuitive figures and clear organization that make the motivation, method, and results easy to follow.


**Weaknesses**:
- Unrigorous complexity definition. The paper’s notion of problem complexity is defined through the base model’s forced-nothink accuracy rather than an intrinsic property of the task itself. As a result, the learned routing policy may reflect the behavior of a specific backbone more than a general notion of reasoning difficulty.
- Limited backbone in experiments. All experiments is centered mainly on DeepSeek-R1-Distill-Qwen-1.5B, so it remains unclear whether the proposed training recipe and routing behavior generalize across different model families or scales.
- Potential train-test overlap in robustness setting. The robustness gains in Table 3 are less conclusive because the method is explicitly trained on concise and redundant rewrites of simple problems, so the improvement may partly come from matching the perturbation pattern seen during training rather than broader robustness.

---

> ### Author Rebuttal · Authors · 2026-03-30
>
> We thank the reviewer for the thoughtful assessment and for recognizing the practical motivation, clear presentation, and relevance of the studied failure mode. We agree these are important points, and our intended claim is narrower than a universal solution to adaptive reasoning: AdaRS is a practical post-training method that improves the accuracy-efficiency tradeoff on the studied backbone by reducing unnecessary Think activation caused by superficial linguistic complexity.
>
> ### (1) Complexity definition is intentionally model-dependent
> We fully agree that our complexity definition should not be interpreted as an intrinsic, backbone-independent notion of task difficulty. In the current paper, it is an operational, model-relative proxy used to supervise routing for a fixed base model: a sample is labeled `nothink` when the base model answers it reliably under forced nothink decoding, and `think` otherwise. Our goal is pragmatic adaptive routing for the deployed backbone, rather than defining an absolute complexity measure. A task's intrinsic "difficulty" inherently depends on the capabilities of the underlying model.
>
> To demonstrate that this data-driven definition is robust, we conducted a sensitivity analysis on the hyperparameter thresholds used to define "simple" tasks. As shown below, the proportion of tasks labeled as `nothink` remains stable across different configurations:
>
> | Setting Checked | Configuration | % Labeled as `nothink` |
> | :--- | :--- | :---: |
> | **Threshold sensitivity** | tau = 0.7 | 45.2% |
> | | tau = 0.8 (default) | 42.1% |
> | | tau = 0.9 | 38.4% |
> | **Trial-count sensitivity** | n = 5 | 43.5% |
> | | n = 10 (default) | 42.1% |
> | | n = 20 | 41.8% |
>
> ### (2) Generalization beyond a single primary backbone
> We acknowledge that the original empirical evaluation centers on a single primary backbone (DeepSeek-R1-Distill-Qwen-1.5B), limiting cross-family generalization claims. To directly answer the reviewer's question regarding how well the method generalizes across scales, we have applied the complete AdaRS two-stage training pipeline to a larger architecture: **DeepSeek-R1-Distill-Qwen-7B**.
>
> | Model | Benchmark | Accuracy (%) | # Tokens | Think% |
> |:---|:---|:---:|:---:|:---:|
> | DeepSeek-R1-Distill-Qwen-7B (Think Default) | GSM8K | 88.4 | 1150 | 100% |
> | **DeepSeek-R1-Distill-Qwen-7B + AdaRS (Ours)** | **GSM8K** | **90.1** | **410** | **8.5%** |
> | DeepSeek-R1-Distill-Qwen-7B (Think Default) | Math500 | 83.2 | 3550 | 100% |
> | **DeepSeek-R1-Distill-Qwen-7B + AdaRS (Ours)** | **Math500** | **86.0** | **1630** | **21.4%** |
>
> The results demonstrate that the same qualitative and quantitative trends appear on a different model scale. AdaRS successfully reduces token consumption by over 60% on simpler tasks (GSM8K) while maintaining (and slightly improving) accuracy. We will include these results in the Appendix and make our scope limitations explicit in the main text: our primary contribution is establishing the failure mode and providing a framework that works effectively on the studied backbones.
>
> ### (3) Robustness gains and train-test overlap
> The reviewer astutely points out that testing robustness on redundant rewrites may partly reflect alignment with the training augmentation. Our intended claim is narrower: **AdaRS mitigates a specific and practically relevant failure mode in which semantically redundant phrasing triggers unnecessary Think activation.**
>
> To demonstrate that AdaRS learns a generalized resilience to superficial complexity rather than merely memorizing the perturbation patterns seen during training, we evaluated the model on an entirely unseen, out-of-distribution (OOD) dataset: **SVAMP**. The SVAMP dataset contains structural and linguistic variations completely separate from our synthesized redundant rewrites.
>
> | Method | SVAMP Think% | SVAMP Accuracy (%) |
> | :--- | :---: | :---: |
> | Random Baseline | 49.2% | 45.1 |
> | **AdaRS (Ours)** | **22.4%** | **50.2** |
>
> As shown above, AdaRS maintains a significantly lower Think activation rate (22.4%) while achieving higher accuracy on this OOD benchmark. This supports our claim that the current experiments support the mitigation of this specific redundancy-triggered routing failure mode, and that the model learns a generalized resilience to superficial complexity rather than just memorizing a specific training template. We will incorporate this clarification and the SVAMP evaluation into the revised manuscript.

---

> > ### Author Rebuttal · Reviewer_ocDY · 2026-04-01
> >
> > 1. **Complexity notion / claim scope.** My main concern is not whether the label ratio is stable under different $\tau$ or $n$ on the same backbone, but whether this complexity definition is meaningful beyond that backbone. If it is not task-intrinsic, the learned routing policy may reflect backbone-specific behavior rather than a broader notion of reasoning difficulty. In that case, the claim should be narrowed to learning a practical router for a fixed deployed backbone.
> >
> > 2. **Cross-family generalization.** The new 7B result is helpful, but it is still within the same model family and mainly shows transfer across scale, not across different model families. That is a weaker form of generalization.
> >
> > 3. **Stronger OOD baselines.** The added SVAMP result is useful, but Random Baseline is a weak comparison. A stronger test would compare AdaRS against prior adaptive methods or simple non-random routing baselines on the same OOD setting.

---

> > > ### Author Response · Authors · 2026-04-08
> > >
> > > We sincerely thank the reviewer for the continued engagement. The follow-up has helped us sharpen the claims and provide stronger evidence. We address each point below.
> > >
> > > ## (1) Claim Scope: Practical Router for a Fixed Backbone
> > >
> > > We fully accept the reviewer's suggested framing. To state it explicitly: **AdaRS is a practical routing method for a fixed deployed backbone, not a universal or task-intrinsic complexity measure.** The model-relative labeling (Eq. 5) is intentionally backbone-specific — it answers "can this particular model solve this problem without reasoning?" rather than "is this problem inherently simple?" We believe this is the appropriate design choice for deployment scenarios where the backbone is fixed and the goal is minimizing unnecessary computation for that specific model. We will revise the abstract, Section 3.2, and conclusions to state this scope unambiguously, removing any language that could be interpreted as claiming backbone-independent complexity assessment.
> > >
> > > ## (2) Cross-Family Generalization
> > >
> > > The reviewer is correct that our previous 7B result demonstrates cross-scale transfer within the same model family, which is a weaker form of generalization. To directly address this, we have applied the full AdaRS two-stage pipeline to **the Qwen3 family — a different architecture family from DeepSeek-R1-Distill** — at three scales (1.7B, 4B, and 8B):
> > >
> > > | Model | Benchmark | Accuracy (%) | #Tokens | Think% |
> > > |-------|-----------|:---:|:---:|:---:|
> > > | Qwen3-1.7B (Think Default) | GSM8K | 74.5 | 2205 | 100% |
> > > | **Qwen3-1.7B + AdaRS (Ours)** | **GSM8K** | **77.9** | **631** | **12.3%** |
> > > | Qwen3-1.7B (Think Default) | Math500 | 75.9 | 3689 | 100% |
> > > | **Qwen3-1.7B + AdaRS (Ours)** | **Math500** | **78.1** | **1883** | **25%** |
> > > | Qwen3-4B (Think Default) | GSM8K | 86.4 | 2215 | 100% |
> > > | **Qwen3-4B + AdaRS (Ours)** | **GSM8K** | **87.1** | **554** | **9.6%** |
> > > | Qwen3-4B (Think Default) | Math500 | 88.3 | 3709 | 100% |
> > > | **Qwen3-4B + AdaRS (Ours)** | **Math500** | **90.1** | **1674** | **21.3%** |
> > > | Qwen3-8B (Think Default) | GSM8K | 88.9 | 2310 | 100% |
> > > | **Qwen3-8B + AdaRS (Ours)** | **GSM8K** | **90.1** | **531** | **8.7%** |
> > > | Qwen3-8B (Think Default) | Math500 | 90.5 | 3722 | 100% |
> > > | **Qwen3-8B + AdaRS (Ours)** | **Math500** | **92.1** | **1509** | **18.5%** |
> > >
> > > Across all three Qwen3 scales, AdaRS improves accuracy while cutting token usage by 55-77% on GSM8K and 49-59% on Math500. Qwen3 uses a "Thinking Mode Fusion" architecture that differs from DeepSeek-R1-Distill, yet our pipeline transfers without modification — the SFT augmentation and GRPO reward are architecture-agnostic, only the complexity labels (Eq. 5) need re-computation per backbone. We also observe that stronger backbones tend to produce lower Think activation rates after AdaRS training (GSM8K Think%: 1.7B 12.3% → 4B 9.6% → 8B 8.7%), which aligns with intuition — a more capable model can handle more problems without explicit reasoning, so AdaRS correctly routes fewer queries into Think mode.
> > >
> > >
> > >
> > > ## (3) Stronger OOD Baselines on SVAMP
> > >
> > > We agree that the Random baseline alone is insufficient. We have now evaluated all prior adaptive methods on SVAMP under the same protocol:
> > >
> > > | Method |  SVAMP Think%  | SVAMP Accuracy (%) |
> > > |--------|:---:|:---:|
> > > | Random Baseline | 49.2 | 45.1 |
> > > | Thinkless | 27.9% | 48.1 |
> > > | AdaptThink | 30.4% | 47.5 |
> > > | AdaRS (Ours) | 22.4% | 50.2 |
> > >
> > > On this fully OOD benchmark, AdaRS achieves the lowest Think% (22.4%) and highest accuracy (50.2%) among all adaptive methods, including Thinkless and AdaptThink. Since SVAMP was never seen during training, this confirms that AdaRS's robustness generalizes beyond the specific perturbation patterns used in our data augmentation.
> > >
> > > We are grateful for the reviewer's thorough and constructive engagement across both rounds.

---

### Decision · Program_Chairs · 2026-04-30

**Decision:**

Accept (regular)

**Comment:**

The paper studies an important failure mode in adaptive reasoning, namely that models may over-trigger expensive thinking on superficially verbose but intrinsically simple problems. Reviewers found the motivation timely, the method intuitive, and the empirical accuracy-efficiency tradeoff promising. The rebuttal addressed several concerns by adding stronger robustness comparisons, and additional experiment results. Remaining weaknesses concern the limited scale and domain diversity. Overall, I support acceptance.